# Plant-Endophyte Interaction during Biotic Stress Management

**DOI:** 10.3390/plants11172203

**Published:** 2022-08-25

**Authors:** Parul Pathak, Vineet Kumar Rai, Hasan Can, Sandeep Kumar Singh, Dharmendra Kumar, Nikunj Bhardwaj, Rajib Roychowdhury, Lucas Carvalho Basilio de Azevedo, Hariom Verma, Ajay Kumar

**Affiliations:** 1Department of Microbiology, Singhania University, Pacheri Bari, Jhunjhunu 333515, India; 2Sri Sudrishti Baba Post Graduate College, Jananayak Chandrashekhar University, Sudishtpuri, Raniganj, Ballia 277208, India; 3Department of Field Crops, Eregli Faculty of Agriculture, Necmettin Erbakan University, Konya 42130, Turkey; 4Division of Microbiology, Indian Agricultural Research Institute, Pusa, New Delhi 110012, India; 5Centre of Advanced Study in Botany, Institute of Science, Banaras Hindu University, Varanasi 221005, India; 6Department of Zoology, Maharaj Singh College, Maa Shakumbhari University, Saharanpur 247001, India; 7Department of Plant Pathology and Weed Research, Institute of Plant Protection, Agricultural Research Organization (ARO)—Volcani Center, Bet Dagan 50250, Israel; 8Instituto de Ciências Agrárias, Campus Glória-Bloco CCG, Universidade Federal de Uberlândia, Rodovia BR-050, KM 78, S/N, Uberlândia CEP 38410-337, Brazil; 9Department of Zoology, Pachhunga University College Campus, Mizoram University (A Central University), Aizawl 796001, India; 10Department of Botany, B.R.D. Government Degree College Duddhi, Sonbhadra 231208, India; 11Department of Postharvest Science, A.R.O., Volcani Center, Bet Dagan 50250, Israel

**Keywords:** endophyte, plant-microbe interaction, molecular aspect of colonization, biotic stress management

## Abstract

Plants interact with diverse microbial communities and share complex relationships with each other. The intimate association between microbes and their host mutually benefit each other and provide stability against various biotic and abiotic stresses to plants. Endophytes are heterogeneous groups of microbes that live inside the host tissue without showing any apparent sign of infection. However, their functional attributes such as nutrient acquisition, phytohormone modulation, synthesis of bioactive compounds, and antioxidant enzymes of endophytes are similar to the other rhizospheric microorganisms. Nevertheless, their higher colonization efficacy and stability against abiotic stress make them superior to other microorganisms. In recent studies, the potential role of endophytes in bioprospecting has been broadly reported. However, the molecular aspect of host–endophyte interactions is still unclear. In this study, we have briefly discussed the endophyte biology, colonization efficacy and diversity pattern of endophytes. In addition, it also summarizes the molecular aspect of plant–endophyte interaction in biotic stress management.

## 1. Introduction

The biology of endophytic microorganisms has been gaining momentum in the last few years due to better colonization efficacy and acclimatization potential against biotic and abiotic stress. In the last few years, endophytes, bacteria, or fungi have been frequently applied in sustainable agricultural practices as biofertilizers to meet nutrient requirements. Biocontrol agents have been used to prevent pathogen invasion or disease control and to mitigate various abiotic stresses, including salinity, drought, etc. The prospect of an endophytic microbiome has been reported in various review papers, which have been published recently [1,2], while molecular aspects of plant–endophyte interactions have been not covered extensively [3].

Plant-microbe interaction is a complex process in which the plant system interacts with diverse heterotrophic microorganisms and can share an intimate relationship from symbiotism to parasitism [4,5,6]. The intimate association of microbes with plants has a long history, and it is assumed that both have co-evolved together since the time of plants’ origin [7]. This inseparable relationship is also referred to as second genomes or holobionts, which play a significant role in maintaining plant health and fitness [8]. The term holobionts is also used as collective term for the microbiome associated with the host and referred as a single entity, which provides genomic reflection and stability to plants under various biotic and abiotic stresses [9]. The functional attributes of a plant as it secretes a range of sugars, metabolites organic or volatiles compounds are also dependent on the associated microorganism. Still, their exact mechanism is unclear at the community level. However, the study of synthetic communities and their outcome can be used to explore the colonization and assembly pattern of microorganism, which can be used to control pathogen invasion and biotic stress management [10,11]. The interaction of plants with microbes is mediated through various organic metabolites or signalling molecules. Their secretions include organic compounds such as amino acids, lipids, polysaccharides, flavonoids etc., that attract the microbial strains for colonization. For example, Steinkellner et al. [12] reported the functional role of root exudates, flavonoids, and strigolactones in the root colonization and hyphal growth differentiation of various *Fusarium* species and also their role as signaling molecules during symbiotic and pathogenic plant-fungus interactions. Similarly, Oku et al. [13] reported the role of amino acids in the root colonization of *Pseudomonas fluorescens* Pf0-1 to tomato plants.

The plant’s rhizosphere is a hot spot of microbial communities and is considered as the favourable site of plant-microbe interaction due to its abundantly present root exudates. Their composition depends upon plant genotypes, development stages, and the surrounding environmental conditions of the rhizospheric microbiota [14]. However, some of the rhizospheric microbes enter the host tissue through natural openings such as stomata, pores, wounds, and hydathodes, acting as endophytes. The entry and establishment of endophytes is a complex process accomplished by various signalling molecules and colonization processes.

In this review article we have briefly discussed the endophyte biology, colonization efficacy and diversity pattern of endophytes. In addition, we also summarize the molecular aspect of plant–endophyte interaction during biotic stress management.

## 2. An Overview of Microbial Endophytes

Endophytes are a heterogeneous group of microbes that live inside the host tissue without showing any external signs of infections. De Bary [15] firstly used the term endophytes for the microbes residing in the host tissue. However, later authors modified the concept and defined endophytes per their observation. Now, in general, any microbes living inside the host tissue for at least a part of their life cycle are considered an “endophyte”, and every plant species have some endophytes in their life cycle [16,17]. However, with the advent of the latest forms of omics and various kinds of technology, extensive research has been carried out on the endophytic microbiome. Initially, only cultivable microbial species isolated from the surface-sterilized tissue have been screened as endophytes. However, the latest next-generation sequencing technology frequently reported the endophytic microbiome of any tissue or host plant, including the non-cultivable microbial species even present in a small number of samples.

Endophytes comprise diverse microbial communities, including various groups of fungi, bacteria, actinomycetes, etc. These endophytic microbial communities classified into different groups based on the host specificity, colonization pattern, transmission mechanism, and evolutionary relationship [18]. The most common fungal endophytes have been generally comprised of two groups: clavicipitaceous, comprising the endophytes harbored by grasses, and non-clavicipataceous, which colonize angiosperm, gymnosperm, and nonvascular plants. However, *Epichloe* (formely *Neotyphodium*), *Claviceps* (Clavicipitaceae), *Cladosporium* (Cladosporiaceae), *Colletotrichum* (Glomerellaceae), *Piriformospora* (Sebacinaceae), *Stemphylium* (Pleosporaceae), *Acremonium* and *Trichoderma* (Hypocreaceae) are the some common genera, and Glomeromycota followed by Ascomycota and Basidiomycota are the dominant endophytic fungal phyla. *Pseudomonas*, *Bacıllus*, *Acinetobacter*, *Brevibacterium*, and *Rhizobium* are the dominant bacterial genera, and *Proteobacteria* followed by *Actinobacteria*, *Firmicutes*, and *Bacteroides* are the dominant bacterial phyla reported as endophytes in most of the plant species [19].

## 3. Entry and Transmission of Endophytes into Plant Tissue

The colonization of endophytic microbial strain to the host tissue is a series of consecutive events mediated by various secretory products and signalling molecules. However, attaching microbial strains to the host tissue is a primary step for endophyte colonization. The secretory product, such as lipopolysaccharides, and exopolysaccharides, directly or indirectly help the adhesion. In contrast, the structural components such as flagella, fimbriae, and pilli help in the movement of bacterial strains towards the host tissue during colonization. However, plants respond differentially after attachments of microbial strains to their surface especially in the gene expression patterns. In a study Bodilis and Barray [20] discussed the outer membrane porin F (OprF) proteins present on the surface of *Pseudomonas* and their role in the attachment with the host surfaces. Similarly the arabinogalactan proteins present on the plant cell wall help in the initial colonization of endophytic microorganisms [21]. After attachment, microbial strains start penetrating to enter the host tissue, which can be mediated through either active or passive processes. The passive penetration taken part at the cracks present on the site of root surface caused by the deleterious organism, while active penetration involved the structural components and secretory products in the entry or multiplication inside the host tissue [22].

The aerial parts such as the stomata, wounds, and cotyledons are the typical entry sites of endophytes [23]. However, significant differences between mutualistic and pathogenic strains have been observed during the process of entry and penetration. For example, the pathogens secrete higher amount of cell wall degrading enzymes in compared to the mutualistic microorganisms [24]. Therefore, mutualistic endophytes and hosts play a deeper and more precise modulation of molecular signalling compared to the pathogens during colonization of plant tissues.

The transmission of endophytes within the host may be through different modes (e.g., horizontal, which is mediated through environments; vertical, referred to as transmission via parents to the offspring through seeds/pollen grains; or mixed ones, which either follows the horizontal and vertical or both modes of transmission) [25]. In a previous study, vertical modes of transmission were well documented in the fungal strains through the isolation of endophytes from seeds, cotyledons and leaves of forb species that grew under sterile conditions [26]. However, the horizontal transmission mode in fungal species has also been reported by various authors. This observation is based on some ubiquitous fungal sporophytes such as *Alternaria* and *Cladosporium*, which generally sporulate on the dead leaf and soil but are frequently reported as endophytes [27,28]. Similarly, bacterial species generally prefer the horizontal, vertical and mixed types of transmission. In another study, the author reported that a seed growing under aseptic conditions has lower bacterial diversity than one grown under normal environmental conditions. This observation suggests the prevalence of the horizontal transmission mode [29,30,31].

The successful colonization of endophytes depends upon several factors such as host genotypes, the nature of microbial strains, and the availability of nutrient sources [32,33,34]. The endophytes, during colonization and transmission, followed an unique path. For example, the strain *Paraburkholderia phytofirmans* PsJN made an entry to the host tissue through the exodermis or cortical cells and reached to the central zone of the roots and moved towards the upper zone or above ground tissue by the xylem vessels [35]. However, endophytic microorganisms commonly prefer unsuberized cells to enter the apical root [36]. A detailed overview of endophyte biology and their mode of transmission is discussed in Figure 1.

Although every plant species has at least some endophytes in their life cycle nevertheless their density varies among the plant species or even different organs of the same plants. In general, the diversity of endophytic microorganisms decreases after moving upwards or from the root to the fruits. Therefore, endophytes’ density is lower in the flowers or seeds than in other plant organs such as the roots, stems, and leaves. However, in the case of the other reproductive organs of the plants, higher density and diversity was recorded in the flowers compared to fruits and seeds [32]. Thus, the endophyte diversity is regressive from the roots to the upper part and reduced through the reproductive organs of the plants.

## 4. Molecular Aspect of Plant-Endophyte Interaction Related to Plant Defense

Endophytes have to face several challenges during their growth and survival inside hosts. First, plant hosts and endophytes have forged a complex mutual relationship due to their co-existence and evolution [37,38]. On the one side, thousands of years of co-evolution process have forced both plants and endophytes to shape their genome to survive together. On the other side, this process has enabled plants and endophytes to develop complex biochemical mechanisms for communicating the chemical weapons against pathogens. Their increased capacity to induce resistance against diseases and unique secondary metabolites produced by endophytes, due to their co-evolution with hosts and horizontal gene transfer (HGT), provide them a greater advantage compared to the conventional microorganisms as biocontrol agents. The non-pathogenic complex relationship between host and the endophytes is explained by the “balanced antagonism” term, which means that both hosts and endophytes avoid activating their defence system and toxic metabolite production, respectively [39], which can be beneficial in disease management. As a result of this directed evolutionary process, endophytic bacteria and fungi have developed some specific genes that exhibit endophytic behaviour and acquired the ability to complex connections with host plants [40,41,42]. Nowadays, various fungal groups including Trichoderma, Fusarium and Phoma, yeast-like Aureobasidium, Merozyma, Metschnikowia, Cryptococcus, Saccharomyces and bacterial groups such as Pseudomonas, Bacillus, Pantoea, etc. have demonstrated some common defensive responses against the pathogens such as antibiosis, lytic enzyme, parasitism and competition, siderophore production, and indirect responses by inducing ISR (Induced Systemic Resistance) or SAR (SystemicAcquired Resistance) of the plant [43,44]. In the phyllosphere, the specific type of bacteria of genera Methylobacterium, Sphingomonas, and Pseudomonas could colonize successfully and have the biocontrol mechanism above ground-based plant pathogens [45,46].

### Interplay between the Endophytes and the Plant Defence against Pathogens

For initial contact, plants recruit endophytes for the colonization to the surface and support/cope against biotic and abiotic stress by exudating various metabolites, including coumarins, triterpenes, camalexin, flavonoids, and, importantly, strigolactones [47,48,49,50,51,52,53]. The exudates released from the plant roots (particularly flavonoids) are sensed by bacteria using phosphotransferase system or periplasmic binding proteins [54,55]. In rhizobial symbiosis, luteolin, quercetin, kaempferol, myricetin, and genistin are common flavonoids that can act as communication agents from legumes to the bacteria [56]. However, in the bacteria, only those having flagella, pili, or special proteins such as hemagglutinins and curli formations initiate the symbiotic attachment to the root surface and motility for the entry site [20,57]. The attachment of bacterial strains to the root surface is stabilized by some specific polysaccharides (lipopolysaccharide and exopolysaccharides including succinoglycan, rhamnose, the outer membrane lipoprotein, and muropeptide permease) that has been described in detail by Pinski et al. [58]. However after attachment and biofilm formation, some endophytic bacteria have the capability of modifying plant cell walls via degrading enzymes (CWDEs) such as cellulases, xylanases, pectinases, endoglucanases, and (rarely) expansions of the metabolism (i.e., cellular growth using Arabinogalactan proteins as explained by Nguema-Ona et al. [59] for successful symbiotic relations.

During this process, microbe-associated molecular patterns (MAMPs) molecules such as flagellin, lipopolysaccharides (LPS), bacterial cold shock proteins (RNP1motif), bacterial superoxide dismutase (SOD) (from beneficial bacteria), and Nod-genes derived lipochitooligosaccharides (LCOs), exopolysaccharides (EPS) (from rhizobial bacteria) secrete in response to plant signals [60]. MAMPs have been sensed by the plant using pattern recognition receptors (PRR) and initiate MAMP-triggered immunity (MTI) as a response to signal perception. However, if the bacteria do not have type 3 or 6 secretory systems (T3SS), then recognized as pathogens and the hypersensitive response and systemic acquired resistance (SAR) is activated in the plant using synergistically interaction of SA and JA–ET pathways [61]. However, if the bacteria have T3SS that can surpass MTI, symbiotic relation and cellular reprogramming (nodulation) and induced systemic resistance (ISR) are activated against other microorganisms in the plant using only JA–ET pathways. In a second way, pathogen directly triggers SAR (SA signalling) in the plant through the way of Effector Triggered Immunity (ETI) (generally triggered by symbiotic ones) responses using as T3SS elicitor proteins [62]. Plant defense system responses such as switching on the defense-related genes, degrading/mimicking acyl homoserine lactone (AHL) such as molecules (for blocking quorum sensing of pathogens or inducing beneficial one’s growth), and ISR can cause structural hardening via callose production, the production of phytoalexins and volatile organic compounds (VOCs), and antibiosis initiated against pathogens [63,64,65]. At the end of these interactions, bacterial endophytes have biocontrol effects on pathogens via siderophore production, antibiosis, lipopeptide production, quorum quenching, and phytohormone production [66,67,68].

Fungal endophytes generally initiate directional hyphal growth throughout plants to establish symbiotic relationship as reported by Rozpądek et al. [69] during the study of cross-talk between Arabidopsis thaliana and the endophytic fungus Mucor sp. However during the growth period, fungus also continuously releases cell wall degrading enzymes to the rhizospheric environment. At the same time chitin, β-glucans, cerebrosides, ergosterol, elicitins, cell wall glucans, and Myc-LCO types of MAMPs released by the endophytes are recognized with plant-specific PRR (receptor-like kinases and receptor-like proteins reviewed by Saijo et al. [70]. While the plant immunity system is activated, above-mentioned MAMPs-triggered immunity (MTI), MAMPs and Myc-LCO signaling molecules lead to symbiotic reprogramming in the plant root cells [71,72]. Under pathogen attacks, plant immune system has been also activated by the small secreted proteins called elicitors (Effector triggered immunity) in addition to MTI. To suppress the MTI triggered by MAMPs, some small secreted effector protein [73], or most probable Myc-LCO (Nod factors have this ability in the bacteria) molecules evolved in this direction. A growing number of evidence stated that the non-expressor of pathogenesis-related genes 1 (NPR1) is a necessity for the activation of SAR and ISR in the plants [74]. Right after MAMPs signal perception, an intracellular messenger of mitogen-activated protein kinase (MAPK) activates MTI and ETI based immune responses and interacts with NPR1 in plants. Despite MTI and ETI suppression by symbiotic fungi, NPR1 continues its mission and regulates some transcription factors inducing ISR [74,75]. Fungal endophytes fight against pathogens either directly through producing antibiotic, various metabolites, establishing a nutrient delivery system for the plants, producing CWDEs and toxins, absorbing nutrients of pathogens, or indirectly triggering plant immune responses by strengthening the structure of entry points, producing reactive oxygen species (ROS) and other reactive oxygens (called oxidative burst), producing pathogenesis-related protein and producing various metabolite [43,76].

In the biocontrol process, the endophyte’s directional growth towards the pathogen is governed by the perception of some unknown released protein from pathogens or degraded cell wall components through CWDEs [71]. After first contact, CWDEs and peptaibols, as with antibiotic molecules, are massively synthesized by the endophytic fungus during coiling formation and also serve in pathogen counterattacks (pathogenic CWDEs) [72].

In the phyllosphere, various types of bacteria, fungi, and actinobacteria have gained unique habitat adaptation to survive against biotic stress, low nutrient condition, water scarcity, electromagnetic radiation, and antimicrobial and toxic substances [77,78,79]. As with the rhizosphere, plant leaf also exudates (VOCs) or leach nutrients (sugar, amino acids, organic acids, etc.) from specified structures such as hydathodes and glandular trichome for preferentially shaping the leaf microbiota [80,81]. Furthermore, aboveground endophytes could directly inhibit pathogens through producing antimicrobial metabolites, hydrolytic enzymes, quorum sensing/quenching, siderophores, competing for nutrients and space, or indirectly by regulating plant immune and hormone system (Legein et al., 2020). In addition to MTI and ETI, microbial sensing systems of plants, plants, and endophytes can sense AHLs and oligopeptide autoinducers which are coming from leaves surface localized bacterial pathogens.

Endophytes suppress the plant defense system in the leaves with similar mechanism, which are involved in the root system. Endophytic microbiota produces direct responses such as AHL-acylase, AHL lactonase, and polyhydroxyanthraquinones for blocking these pathogenic QS signals in the phyllosphere for biocontrol purposes [82]. In addition to the production of antimicrobial metabolites, hydrolytic enzymes, siderophores, and competing for nutrients and space represent the common biocontrol mechanisms [83,84]. An example of properties acquired during horizontal gene transfer in the evolutionary process, endophytic ACC deaminase activity acts as a signaling molecules and forms conjugates for cross-communicating with jasmonic acid regulation to the plant defense system against the pathogens [85]. In order to form successful colonization between plants and endophytes, there should be low or even no ethylene (ET) accumulation should be happened in the host tissue, and this process have been enabled by ACC deaminase activity [86]. Since the accumulation of ethylene, during the pathogenic attack, causes inhibition of plant growth by regulating ROS, hydroxyproline-rich glycoproteins (HPRGs), plasma membrane H^+^-ATPases, and some other cell wall remodeling enzymes, which also lowered the endophytic colonization either in the rhizosphere or phyllosphere. For this reason, endophytes have a negative impact on the ET production, either directly through producing ACC deaminase, which inhibit ET accumulation or indirectly via inhibiting ACS (1-aminocyclopropane-1-carboxylate synthase) expression by producing vinylglycine analogue [86]. Thus, during this process, plants’ growth or development process with the help of ACC deaminase-based ET decomposition act as a bridge during pathogen attacks and mutualistic association [87].

However, knowing the molecular consequences of endophytic colonization on the host surface and interiors during their interactions with pathogens is imperative. Numerous pieces of literature on the bipartite interaction between pathogens and host describes the metabolic and molecular alteration in the host plants. However, not many studies have reported the tripartite interaction of host pathogens and endophytes on the molecular and metabolic scale. For example, jasmonate-salicylate crosstalk and ethylene signalling play a vital role in interacting with the host plant and biotrophic pathogens [88]. Similarly, adding bioactive compounds or biocontrol agents to the fruits is known to accumulate pathogenesis-related (PR) proteins, activate hormone-dependent pathways, and elicit antioxidant machinery while pathogen attack [89]. The molecular and metabolic changes occurring in the host plant during the tripartite interaction of host-pathogen-endophytes are of great concern regarding preserving the nature, texture, and nutritional values [90]. The plant responses during tripartite interaction of plant-pathogen and beneficial microbes in the field conditions can also provide a path for studying molecular responses during simultaneous interactions with pathogens and endophytes. Kumari et al. [91] deciphered the proteomic and metabolomic alterations in the host plant Arabidopsis thaliana during interactions with the pathogen Alternaria brasiccicola and nanoparticles synthesized from biocontrol agents (e.g., Trichoderma viride) in an omics-based study. The interaction conferred biotic stress tolerance and increased defence responses to the host plants. Fiji and Babalola [92] elucidated the multifaceted role of endophytes in plant protection and metabolic alterations.

## 5. The Potential Application of Endophytes in Phytopathogen Management

In recent years, microbial biocontrol agents have been frequently employed to control the pests and pathogens during pre or post-harvest storage conditions of crops [93]. The non-toxic and economic nature of microbial antagonistic makes them popular globally. Available chemical pesticides, which are widely used by farmers and horticulturists in the field or in storage not only affect the natural texture and quality of the crops/fruits but also adversly affect consumer’s health. In this regard, utilising microbial antagonistic to control pest or disease management has been gaining momentum in last two decades [43]. However, the utilization of endophytic microorganisms as biocontrol agents have is more advantageous over other microbial species due to their better colonization efficacy and acclimatization potential under stress conditions [94]. The biocontrol mechanism of endophytes is very similar to other rhizospheric microorganisms such as mycoparasitism, competition for nutrition and space, production of lytic enzymes, synthesis of bioactive compounds, antibiotics, antimicrobial and volatiles compounds, which directly inhibit phytopathogen growth and suppress disease incidence. At the same time, indirectly elicit the local and systemic plant defence systems of the host plants and protect from pathogen invasion [93,95].

Recently, various authors successfully utilized the endophytic strain in phytopathogen control. De Silva and coauthors [96] extensively review the biocontrol potential of endophytic fungi and their potential role in plant disease management. Chen et al. [97] reported biocontrol potential of endophytic strain Lactobacillus Plantarum, which effectively inhibits the mycelial growth of Botrytis cinerea in both in-vitro and in-vivo study. Sudhir et al. [98] utilized endophytic bacterial strain Pseudomonas aeruginosa to control anthracnose disease in chilli. Madbouly et al. [99] successfully utilized different endophytic yeast strains to control pathogen Monilinia fructigena. İn another study, Xu et al. [100] isolated 608 cultivable endophytic bacterial strains of 36 genera from the different mulberry cultivars, wherein 33 strains exhibited strong biocontrol potential and stable activity against the pathogens Sclerotinia sclerotiorum, Botrytis cinerea, and Colletotrichum gloeosporioide. Similarly, in another study the combined applications of Beauveria and Bacillus effectively reduced wilt disease and fruit borer attack in tomato plants [101]. A detailed summary of endophytic microbial strains used to control the phytopathogen has been described in Table 1.

## 6. Conclusion and Future Perspective

In the last two decades, significant progress has been made in endophytic microbiome research on the utilization of biocontrol agents and biofertilizers. It is known that a mutual and fine tuning of molecular signaling and interactions modulates the colonization of plant tissues by the endophytes. However, a significant fraction of endophytes communities and their functional attributes remain hidden due to unable to culture in the laboratory conditions/culture and not understanding of their molecular mechanism of action. Nowadays, continuous effort has been made to explore the novel biocontrol agent as an alternative to chemical pesticides that not only affect the texture and nutrient quality of fruits and crops, but also adversely affect the health of consumers. Better colonization efficacy and stability against the environmental fluctuations make the endophytes a suitable biocontrol agent compared to another microorganisms. However, the competition for nutrients and space, mycoparasitism, and synthesis of volatiles compounds are the most common mechanisms of biocontrol action. Nevertheless, a significant fraction of endophytes’ functional attributes has been hidden, which can be explored by a deep study of complex plant endophyte interactions. Furthermore, the molecular mechanism of host- endophytes’ pathogen interactions can provide a broad understanding for biocontrol screening and its successful application.

## Figures and Tables

**Figure 1 plants-11-02203-f001:**
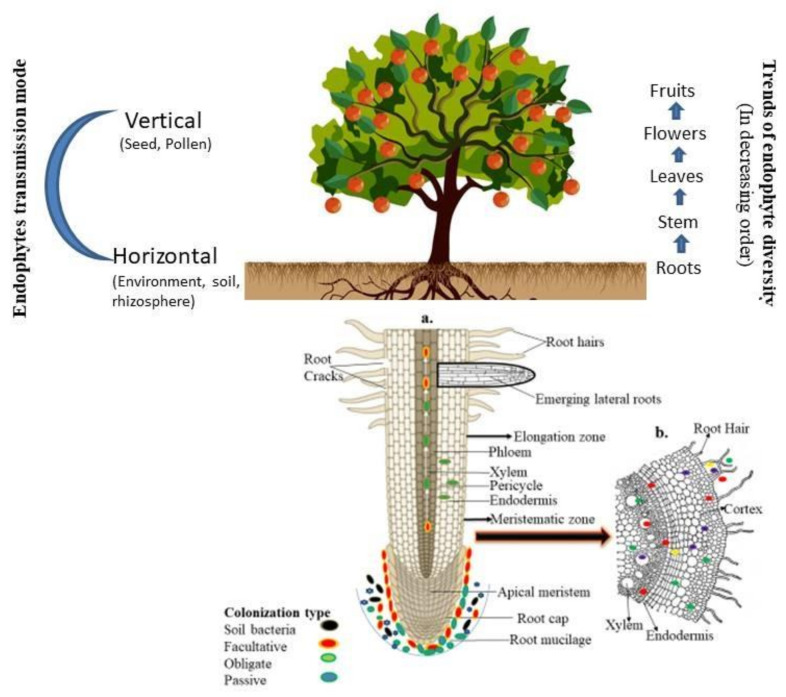
The detailed overview of endophyte biology and their mode of transmission: (**a**) entry of microbial endophytes to the plant tissue by different part of root zones. Arrows show the movement of endophytes; (**b**) occurrence of endophytes either at the entry site or in the intercellular spaces. Part of a figure has been taken from Kumar et al. [32].

**Table 1 plants-11-02203-t001:** Biocontrol agents and their mechanism of action against pathogenic microorganism and their associated crops.

Biocontrol Microorganism	Pathogen	Mechanism	Fruit/Crop	References
*Bacillus subtilis* 7PJ-16	*Sclerotiniose*	Antibiosis	Mulberry	[102]
*Bacillus subtilis* GLB191	*Plasmopara viticola*	Antibiosis	Grapevine	[103]
*Alcaligenes faecalis* subsp. faecalis str. S8	*Fusarium* Wilt	Antibiosis, chitinolytic, proteolytic and pectinolytic enzymes and hydrogen cyanide.	Tomato	[104]
*Bacillus* sp.	*F. Avenaciarum*; *F. sambucinum*; *F. Oxysporum*	In vitro antibiosis	Potato	[105]
*Bacillus pumilus* SE34	*F. oxysporum* f. sp. pisi	Epidermal strengthening and cortical cell wall	Pea	[106]
*Trichoderma* sp. Strain T154	*Phaeoacremonium minimum*	Antibiosis	*Vitis vinifera* L.	[107]
*Bacillus tequilensis* GYLH001	*Magnaporthe oryzae*	cellulase, protease, gelatinase, indole-3-acetic acid and 1-amino-cyclopropane-1-carboxylate deaminase	Rice	[108]
*Pseudomonas putida*	*Fusarium* wilt by *Fusarium oxysporum* f. sp. melonis	Antibiosis	*Cucumis melo*	[109]
*Trichoderma harzianum*	Brown spot by *Bipolaris oryzae*	Antibiosis, increase total photosynthetic pigment	Rice	[110]

## Data Availability

Not applicable.

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
