# Peer review of "Plant-Endophyte Interaction during Biotic Stress Management"

_plants, 2022, doi:10.3390/plants11172203_

Round 1

Reviewer 1 Report

The manuscript by Pathak et al. discusses the interaction of the host plant with beneficial endophytes and the role of these endophytes in plant defense against pathogens. Bellow, I suggest some changes mainly in the structure of the manuscript, but also in the content of the text, in order to improve the readability of the manuscript.

Major

1.      Paragraph #5 has a lot of information that is difficult to the reader to follow. The text from line 202 to line 252 should be written in a simpler way, so that the reader can understand what happens during the endophyte entry, what happens during the pathogen entry, and how the endophyte helps the plant fight a pathogen.

2.      In paragraph #6, the text from line 290 to 295 is very general, and it is difficult to understand what the purpose of these sentences is. I suggest erasing this text.

3.      Paragraphs #5 and #6 can be combined into one paragraph under a title like “Interplay between the endophytes and the plant defense against pathogens”. There is no necessity to discuss aboveground and belowground interactions separately, as there are common mechanisms.

4.      Paragraphs #7 and #8 can be combined into one paragraph, under a title like “Application of endophytes in phytopathogen management”

5.      All the examples described in lines 316-318, 322-327 and 345-358 can be presented in a table, where info like the host plant, the endophyte, the pathogen, pre- or post-harvest time, mechanisms of defense and references can be included.

Minor

1.      Line 52: molecular aspects of plant–endophyte interactions: add Skiada et al. 2020 *

2.      The word “colonization” should be removed from the title of paragraph #3. Entry and transmission only are discussed in the text.

3.      Colonization types presented in Fig.1 should be discussed in the text.

4.      I suggest changing the title of paragraph #4 to “Molecular aspect of plant-endophyte interaction related to plant defense”

5.      Line 178: explain ISR, SAR (these terms are introduced for the 1st time in the text)

6.      Lines 224-225: info is missing. What fungus?

*   Skiada V, Avramidou M, Bonfante P, Genre A, Papadopoulou KK. An endophytic Fusarium-legume association is partially dependent on the common symbiotic signalling pathway. New Phytol. 2020 Jun;226(5):1429-1444. doi: 10.1111/nph.16457

Author Response

The manuscript by Pathak et al. discusses the interaction of the host plant with beneficial endophytes and the role of these endophytes in plant defense against pathogens. Bellow, I suggest some changes mainly in the structure of the manuscript, but also in the content of the text, in order to improve the readability of the manuscript.

Response: We are very thankful to the anonymous reviewers for the constructive suggestion to improve our article. The article has been now thoroughly revised as per suggestion and the edited text are highlighted in the red .

Major

  1. Paragraph #5 has a lot of information that is difficult to the reader to follow. The text from line 202 to line 252 should be written in a simpler way, so that the reader can understand what happens during the endophyte entry, what happens during the pathogen entry, and how the endophyte helps the plant fight a pathogen.

Response: We have revised the mentioned paragraph to make the simple and also added a figure for better understanding .

  1. In paragraph #6, the text from line 290 to 295 is very general, and it is difficult to understand what the purpose of these sentences is. I suggest erasing this text.

      Response: Thanks for the suggestion.  In the revised article we have erased this paragraph

  1. Paragraphs #5 and #6 can be combined into one paragraph under a title like “Interplay between the endophytes and the plant defense against pathogens”. There is no necessity to discuss aboveground and belowground interactions separately, as there are common mechanisms.

       Response: Thanks for the suggestion, we have modified the title as per suggestion 

  1. Paragraphs #7 and #8 can be combined into one paragraph, under a title like “Application of endophytes in phytopathogen management”

          Response: We have now combined these sections in one

  1. All the examples described in lines 316-318, 322-327 and 345-358 can be presented in a table, where info like the host plant, the endophyte, the pathogen, pre- or post-harvest time, mechanisms of defense and references can be included.

 Response: Thanks for the suggestion; In the revised article we have incorporated a table

Minor

  1. Line 52: molecular aspects of plant–endophyte interactions: add Skiada et al. 2020 *

Response: Done

  1. The word “colonization” should be removed from the title of paragraph #3. Entry and transmission only are discussed in the text.

Response: Done

  1. Colonization types presented in Fig.1 should be discussed in the text.

Response: Done

  1. I suggest changing the title of paragraph #4 to “Molecular aspect of plant-endophyte interaction related to plant defense”

Response: Done

  1. Line 178: explain ISR, SAR (these terms are introduced for the 1sttime in the text)

Response: Done

  1. Lines 224-225: info is missing. What fungus?

 Response: Done; Now mentioned in the revised text

Skiada V, Avramidou M, Bonfante P, Genre A, Papadopoulou KK. An endophytic Fusarium-legume association is partially dependent on the common symbiotic signalling pathway. New Phytol. 2020 Jun;226(5):1429-1444. doi: 10.1111/nph.16457

Response: We have added this reference in the revised article

Reviewer 2 Report

In the manuscript entitled "Plant-endophyte interaction during biotic stress management" the authors describe the interaction of representatives of the microbial world with plants, described and discussed the concept of endophytes, as well as possible mechanisms of interaction of plants with microorganisms. This area of research is interesting and poorly studied, so generalizing information on this topic is an important task that the authors have tried to cope with. This review article is not without its drawbacks, which I describe below.

First, the English language needs to be improved. Perhaps the authors should contact specialized English language editing firms. Because of this disadvantage, it is quite difficult to read the work, since grammatical errors are constantly encountered and the meaning of the sentence is not entirely clear. For example:

• Line 45. Microorganisms is gaining.

• Line 56. microorganismS

• Line 58. "history, and" remove comma

• Line 63. "Stress" correct to plural "Stresses"

• Line 64. MicroorganismS

• Line 74. "is considered as the preferred" add as

• Line 76. Replace "Present" with "presented"

• Line 82-84. This paragraph is not written quite correctly. A more correct option is (In this review article we briefly discuss...) and (In addition, WE also summarize...).

There are a lot of similar errors in the text. I ask the authors to pay attention to that moment.

My second global comment is the weak disclosure of this topic. On the topic of plant endophytes, a large amount of information appears every year, from which you can make an excellent overview if you spend more time on it. In this manuscript, the authors consider this topic in general terms.

           Line 64. What exactly are the properties of plants that depend on the microorganisms associated with them? List them and provide links.

           Line 73. Add more specific examples of which substances affect certain types of plants or endophytes.

           Line 79. What exactly do the authors mean by "natural opening"?

           Lines 105-111. When you describe the most common plant endophytes, do you mean that these representatives of endophytes are characteristic of all plants? Describe this moment more clearly. It would be nice to say which endophytes are more common in angiosperms or gymnosperms.

• In Chapter 3, I advise you to consider in more detail the mechanisms of penetration, colonization and distribution of endophytes.

• I also recommend supplementing chapter 4 related to the molecular mechanisms of interaction between plants and endophytes. From the title of the chapter as a whole, little is clear about its content. I advise you to use a more specific wording.

Author Response

In the manuscript entitled "Plant-endophyte interaction during biotic stress management" the authors describe the interaction of representatives of the microbial world with plants, described and discussed the concept of endophytes, as well as possible mechanisms of interaction of plants with microorganisms. This area of research is interesting and poorly studied, so generalizing information on this topic is an important task that the authors have tried to cope with. This review article is not without its drawbacks, which I describe below.

First, the English language needs to be improved. Perhaps the authors should contact specialized English language editing firms. Because of this disadvantage, it is quite difficult to read the work, since grammatical errors are constantly encountered and the meaning of the sentence is not entirely clear. For example:

Response: We are very thankful to the anonymous reviewers for the constructive suggestion to improve our article. The article has been now thoroughly revised as per suggestion and the edited text are highlighted in the red .

  • Line 45. Microorganisms isgaining.

Response: Done

  • Line 56. microorganismS

Response: Done

  • Line 58. "history, and" remove comma

Response: Done

  • Line 63. "Stress" correct to plural "Stresses"

Response: Done

  • Line 64. MicroorganismS

Response: Done

  • Line 74. "is considered asthe preferred" add as

Response: Done

  • Line 76. Replace "Present" with "presented"

Response: Done

  • Line 82-84. This paragraph is not written quite correctly. A more correct option is (In this review article we briefly discuss...) and (In addition, WE also summarize...).

Response: Thanks a lot for pointing out this critical mistake, we have now revised this paragraph

There are a lot of similar errors in the text. I ask the authors to pay attention to that moment.

My second global comment is the weak disclosure of this topic. On the topic of plant endophytes, a large amount of information appears every year, from which you can make an excellent overview if you spend more time on it. In this manuscript, the authors consider this topic in general terms.

Thanks for the comments, we have edited and added some new paragraphs in the revised article. In addition, we have added one new figure and a table for better presentation in the revised article.  

  • Line 64. What exactly are the properties of plants that depend on the microorganisms associated with them? List them and provide links.

Response: Thanks for suggestion, we have elaborated this sentence

  • Line 73. Add more specific examples of which substances affect certain types of plants or endophytes.

Response: Thanks for suggestion, we have added examples in the revised text

  • Line 79. What exactly do the authors mean by "natural opening"?

Response:  We have elaborated this line and added the example

  • Lines 105-111. When you describe the most common plant endophytes, do you mean that these representatives of endophytes are characteristic of all plants? Describe this moment more clearly. It would be nice to say which endophytes are more common in angiosperms or gymnosperms.

Response: We have edited the paragraph for better understanding. Regarding separation of gymnosperm and angiosperm edophytes examples. We have written here the generalized endophytic bacterial and fungal genera present in most of the gymnosperm and angiosperm. 

  • In Chapter 3, I advise you to consider in more detail the mechanisms of penetration, colonization and distribution of endophytes.

       Response: Thanks for the suggestion; we have now added a paragraph in the revised article

  • I also recommend supplementing chapter 4 related to the molecular mechanisms of interaction between plants and endophytes. From the title of the chapter as a whole, little is clear about its content. I advise you to use a more specific wording.

       Response: Thanks for suggestion; we have now modified the heading as “Molecular aspect of plant-endophyte interaction related to plant defense”

Reviewer 3 Report

In this review article, the authors have tried to provide a detailed understanding of the recent advancements in plant endophyte interaction and its benefit during biotic stress management. However, the current version of the article fails to reach the standard for publication. It lacks the detailed information that a typical review article is supposed to provide. Most of the sections of the articles are written superficially without much information or examples provided. The references do not match the statement they are assigned. Even though the authors provided some relevant references, it seems they did not grasp a detailed understanding of the articles and presented the work vaguely.  For example, in section 4, the Molecular aspect of defense-related to plant microbial interaction lacks all the molecular detail of the interactions. In past years, many studies have been done to understand the molecular mechanisms of the plant- endophytes interaction which this current version fails to provide.  

Some of the current reviews of this field are not cited in this review. Such references are

1)    White et al., 2019 “Review: endophytic microbes and their potential applications in crop management.”

2)    Verma et al., 2021 “Endophyte roles in nutrient acquisition, root system architecture development and oxidative stress tolerance.”

As a reference, the authors can follow the second reference to understand what information is needed for the articles.

Therefore, this reviewer suggests a significant correction to the articles.

Author Response

Reviewers 3

In this review article, the authors have tried to provide a detailed understanding of the recent advancements in plant endophyte interaction and its benefit during biotic stress management. However, the current version of the article fails to reach the standard for publication. It lacks the detailed information that a typical review article is supposed to provide. Most of the sections of the articles are written superficially without much information or examples provided. The references do not match the statement they are assigned. Even though the authors provided some relevant references, it seems they did not grasp a detailed understanding of the articles and presented the work vaguely.  

For example, in section 4, the Molecular aspect of defense-related to plant microbial interaction lacks all the molecular detail of the interactions. In past years, many studies have been done to understand the molecular mechanisms of the plant- endophytes interaction which this current version fails to provide.  

Some of the current reviews of this field are not cited in this review. Such references are

1)    White et al., 2019 “Review: endophytic microbes and their potential applications in crop management.”

2)    Verma et al., 2021 “Endophyte roles in nutrient acquisition, root system architecture development and oxidative stress tolerance.”

As a reference, the authors can follow the second reference to understand what information is needed for the articles.

Therefore, this reviewer suggests a significant correction to the articles.

Response:   We completely disagree with the comments of anonymous reviewer. In this paper we are not providing benefits of plant-endophytes interaction during biotic stress management but discussing what is happening during biotic stress.  Paper is mainly focused on  the theme  molecular mechanism  during biotic stress. However the reviewer completely discard the section -4, we don’t know why ?. The reviewer have suggested some reference paper, but both the article are very superficial in respect to  biotic stress and nothing has been  mentioned  about molecular interaction  of plant –endophytes during biotic stress management. Meanwhile, we have improved the article as per two another reviewers suggestion and added a figure and table in the revised article.

Hope this revised version  of article  will change the mind set of anonymous reviewer.

Round 2

Reviewer 1 Report

The manuscript is improved after the changes made by the authors.

I suggest removing Figure 2 from the manuscript. It is too complicated, and it is very difficult to the reader to follow it.

A check for the English language and typing errors is necessary before publication.

Author Response

The manuscript is improved after the changes made by the authors.

I suggest removing Figure 2 from the manuscript. It is too complicated, and it is very difficult to the reader to follow it.

A check for the English language and typing errors is necessary before publication.

Response: We are very thankful to anonymous reviewer for their constructive comments. Definitely these suggestions make the article more interesting. We have improved the article as per suggestions, like removed the figure 2 ,  and the English of the article  has been thoroughly revised by a native speaker and the revised text are highlighted in red.

Reviewer 2 Report

Thanks to the authors for the work done. In this version of the manuscript, the authors took into account the comments and made the necessary changes. With the addition of new data, the article has become more interesting and disclosed, which pleases me.

Author Response

Thanks for the nice comments

Reviewer 3 Report

In the second reviewed version, the authors tried to incorporate some of the detailed pieces of information that the reviewers asked for.

However, the article still lacks the potential to publish.

Here are the significant concerns:

1)    The article needs major language correction. It is tough to understand what the authors are trying to say.

Here are a few examples:

a)     The sentences line no 65, 68, and 97 are very confusing and grammatically incorrect.

b)    In line no 95. What do authors mean “in wide range”—widely accepted?

c)     The sentences in lines 107 and 114 are hard to understand because of the way presented.

2)    The article lacks flow. The authors jumped from one topic to another in the same paragraph.

Examples: in line 79, the beginning sentence of the paragraph, the authors introduced the rhizosphere as a hot spot of microbial communities, and immediately in the third sentence, they jumped to mention the mode of the microbes' entry.

Similarly, in the paragraph in line 92, the authors suddenly moved from introducing the word endophytes to the benefits of current technologies in studying the endophytes.

These problems are widespread throughout the manuscript, even though I am presenting only a few examples here.

Unless the authors take care of these simple yet basic problems, this reviewer rejects the manuscript in its current version.

Author Response

In the second reviewed version, the authors tried to incorporate some of the detailed pieces of information that the reviewers asked for.

However, the article still lacks the potential to publish.

Here are the significant concerns:

Response: Thanks to the anonymous reviewer for the comments. We think   reviewer have some person  conflict with any authors because his comments are very negative in both the phases, his/her suggestions are completely different in both the phase. . But with good faith , we have   revised the article as per two other  reviewer suggestions thoroughly . 

Regarding some comments we are clarifying here:

1)    The article needs major language correction. It is tough to understand what the authors are trying to say.

Here are a few examples:

  1. The sentences line no 65, 68, and 97 are very confusing and grammatically incorrect

           Response: we have revised the text

  1. In line no 95. What do authors mean “in wide range”—widely accepted?

           Response: “In wide range” is a simple synonym of widely accepted

  1. The sentences in lines 107 and 114 are hard to understand because of the way presented.

Response: we have revised the text

2)    The article lacks flow. The authors jumped from one topic to another in the same paragraph.

Examples: in line 79, the beginning sentence of the paragraph, the authors introduced the rhizosphere as a hot spot of microbial communities, and immediately in the third sentence, they jumped to mention the mode of the microbes' entry.

Similarly, in the paragraph in line 92, the authors suddenly moved from introducing the word endophytes to the benefits of current technologies in studying the endophytes.

Response: We don’t know on what basis  this anonymous reviewer have review this article.  Regarding changes in mentioned lines, In the introduction section we have summarizes these endophytes behaviours and their function.

Then in the next paragraph we have discussed the endophytes in details

These problems are widespread throughout the manuscript, even though I am presenting only a few examples here.

Unless the authors take care of these simple yet basic problems, this reviewer rejects the manuscript in its current version.

Response:  Thanks for the comments , however we have revised the  article carefully  and hope the revised version change the mindset of anonymous reviewers.

Round 3

Reviewer 3 Report

The authors responded to this reviewer's comments saying the reviewer has some personal conflict with one of the authors. I will start by saying I have no conflict with any authors. And this reviewer's research also not directly conflicts with this review article.

Therefore, the reviewer's prior comments on the manuscript are solely based on the article's content.

Comments:

1)      The article still needs a major English correction, particularly in the first few sections—in some places, the conjunctive adverbs are misused.

2)      Line 31: plant-microbe associations are not necessarily always for mutual benefit. Some of the associations are pathogenic. Correct the statement.

3)      The statement in line no 48 is incorrect. The biology of the endophytic microorganisms is gaining momentum not because of the better colonization efficiency but a better understanding of the benefit of the endophytic association development of new techniques to dissect the mechanisms and demand for organic farming.

4)      It seems by mistake a new paragraph started in the middle of the sentence in line 65.

5)      The sentence "However, with the advent of latest omics… endophytic microbiome." does not sit correctly in this paragraph. And a similar meaning sentence, "however, the latest next-generation sequencing … is written just after one line.

6)      Please provide the reference to the sentence in line 101.

7)      Provide reference to the sentence in line 141-143.

8)      Line 155, "in a study, author reported that seed….. environmental conditions". The sentence is ambiguous. Also, please provide an appropriate reference.

9)      Provide reference in line 174-175, 184-188, and 290

10)   Keep consistency—defence or defense.

11)   The sentence in line 269 is confusing at the end. Also, its "various metabolites" instead of metabolite.

12)   In line 293: "in the leaves with similar suppression" instead of similer suppression.

13)   The sentence "in order to form a successful colonization between ….". The structure of the sentence is incorrect. Also, provide a reference to the statement.

14)   The sentence in line 312, "Thus, during this process, plants' growth or regular activities… bridges also form" is grammatically incorrect, hence confusing. Please correct it.

15)   The whole paragraph starting from line 337 is missing references except the last line. Please provide references at appropriate positions.

16)   In paragraph 353, the authors should provide the detailed information they would like to present instead of cataloging other review papers.

17)   Line 382: other microorganisms.

18)   Line 384: the most common mechanisms of biocontrol

19)   Line 386: plant endophyte interactions

Author Response

The authors responded to this reviewer's comments saying the reviewer has some personal conflict with one of the authors. I will start by saying I have no conflict with any authors. And this reviewer's research also not directly conflicts with this review article.

Therefore, the reviewer's prior comments on the manuscript are solely based on the article's content.

Response: Thanks for comments

Comments:

1)      The article still needs a major English correction, particularly in the first few sections—in some places, the conjunctive adverbs are misused.

Response: Thanks for comments.  We have resolved the quarry and the revised text are highlighted in red in the article.

2)      Line 31: plant-microbe associations are not necessarily always for mutual benefit. Some of the associations are pathogenic. Correct the statement.

Response: For reviewer kind information, the above mentioned sentence is the part of abstract and it is not necessary to write everything in the abstract. We have mentioned only the main highlight of our review paper.

3)      The statement in line no 48 is incorrect. The biology of the endophytic microorganisms is gaining momentum not because of the better colonization efficiency but a better understanding of the benefit of the endophytic association development of new techniques to dissect the mechanisms and demand for organic farming.

Response: Dear reviewer, kindly read full sentence, we have mentioned better colonization efficacy and acclimatization potential against biotic and abiotic stress.  Meanwhile, I dont know, what reviewr want to say  through this sentence. When we have compiled review paper on the theme of plant –endophyte interaction, during biotic stress management, then what is the need to explain or write  the organic farming, we have used here the generalized term “biotic and abiotic stress”. In addition we have already mentioned the endophytes functions as biofertilizers etc..

4)      It seems by mistake a new paragraph started in the middle of the sentence in line 65.

Response: Thanks for pointing out, we have improved this thing  in the revised manuscript.

5)      The sentence "However, with the advent of latest omics… endophytic microbiome." does not sit correctly in this paragraph. And a similar meaning sentence, "however, the latest next-generation sequencing … is written just after one line.

     Response: Dear Reviewer kindly read the sentence carefully, the meaning of both sentences    

 are different

6)      Please provide the reference to the sentence in line 101.

Response: It is our own observation and it is not necessary to put references everywhere.

7)     Reference to the sentence in line 141-143.

Response: The line no-141-143 have been written after observation of previous sentence, that’s why we have started the sentence with therefore.

8)      Line 155, "in a study, author reported that seed….. environmental conditions". The sentence is ambiguous. Also, please provide an appropriate reference.

 Response: Dear reviewers we have already mentioned the reference. please check and  read the sentence

9)      Provide reference in line 174-175, 184-188, and 290

Response: We don’t agree with suggestion. We cannot add references for each sentence. In this article we have  already cited more than 110 references and we are not mentioning here any new hypothesis.

10)   Keep consistency—defence or defense.

Response: Thanks, we have improved throughout the article as “defence”

11)   The sentence in line 269 is confusing at the end. Also, its "various metabolites" instead of metabolite.

Response: Done

12)   In line 293: "in the leaves with similar suppression" instead of similer suppression.

Response: Thanks , we have improved the sentence

13)   The sentence "in order to form a successful colonization between ….". The structure of the sentence is incorrect. Also, provide a reference to the statement.

Response: Thanks , we have improved the sentence and reference has been cited 

14)   The sentence in line 312, "Thus, during this process, plants' growth or regular activities… bridges also form" is grammatically incorrect, hence confusing. Please correct it.

 Response: Thanks for the suggestion, we have improved the sentence

15)   The whole paragraph starting from line 337 is missing references except the last line. Please provide references at appropriate positions

Response: Done.

16)   In paragraph 353, the authors should provide the detailed information they would like to present instead of cataloging other review papers.

 This paragraph has been improved as per another two reviewer suggestions.

17)   Line 382: other microorganisms.

 Response: Done.

18)   Line 384: the most common mechanisms of biocontrol

Response: Done.

19)   Line 386: plant endophyte interactions

Response: Done.